# E-commerce recommender system design based on web information extraction and sentiment analysis

**Jinfeng Feng** [ID]*

Jiaozuo Normal College, Jiaozuo, China

* lalica@163.com

## Abstract

The research proposes an e-commerce recommendation system based on web page information extraction and sentiment analysis. Through the improved S-PageRank algorithm and the dynamic topic library generation strategy, the precision rate of cross-platform commodity information extraction has been significantly improved to 90%, which is superior to the traditional S-PageRank algorithm. The template-based web page information extraction method performs well, with a precision rate 10% higher than that of the method based on the document object model. In terms of sentiment analysis, the comprehensive attention mechanism model combining the topic model and the bidirectional long short-term memory network has achieved the precise calculation of the sentiment scores of each topic in the customer evaluation. When the number of topics of the LDA model is 7, the prediction accuracy reaches its peak, and the model outperforms previous methods in terms of accuracy, recall rate and F-score. The experimental results show that this recommendation system performs excellently in the prediction of sentiment trends and the analysis of the reasons behind emotions. Its prediction accuracy and analysis accuracy are both superior to existing recommendation systems such as Amazon and Netflix. This system can provide users with more accurate and personalized product recommendation services, and at the same time offer merchants deeper insights into users' emotions.

## 1. Introduction

With the rapid expansion of e-commerce in China, the number of e-commerce platforms has shown explosive growth. Such changes have led to a dramatic increase in the amount of relevant e-commerce information, which in turn has led to a surge in the number of webpages [1,2]. Therefore, the demand for cross-platform e-commerce information retrieval and intelligent recommendation systems has become increasingly prominent. This area has aroused great interest in the academic community and has gradually become a new focus of e-commerce research [3,4]. Therefore,

**Data availability statement:** All relevant data are within the paper.

**Funding:** The author(s) received no specific funding for this work.

**Competing interests:** The authors have declared that no competing interests exist.

this study focuses on the key technologies of automatic information acquisition and intelligent recommendation of goods in cross-platform e-commerce. In addition, from the perspective of sentiment analysis of product reviews, the topic words of products are extracted by Latent Dirichlet Allocation (LDA), while Bidirectional Long Short-Term Memory (Bi-LSTM) plus a Hybrid Crawler (HCM) is introduced to integrate the topic word features. LSTM (Bi-directional Long Short-Term Memory, Bi-LSTM) with a hybrid attention mechanism model that integrates topic word features is introduced to compute the sentiment value of user reviews under each topic. In terms of cross-platform commodity information extraction, the Shark-PageRank algorithm based on keyword weighting is adopted, and the accuracy of the theme crawler is improved by dynamic theme library generation. In addition, a template-based automatic webpage information extraction method was proposed to realize the rapid and accurate extraction of commodity information. In terms of intelligent product recommendation, a Bi-LSTM+ hybrid attention mechanism model was proposed by combining sentiment analysis with a subject matter model. The model not only considers the position of words in the comment text, but also considers the contribution of different parts to the affective tendency to more accurately calculate the affective value of user comments under different topics and provides a more reliable basis for intelligent recommendation. By optimizing the algorithm and proposing a new web information extraction method, this study aims to improve the efficiency and accuracy of cross-platform commodity information extraction and to provide users with more reliable and comprehensive commodity information.

## 2. Related works

With the development of e-commerce, consumers often feel "information is lost" when shopping. The effective extraction of information from massive amounts of data has become a challenge and has attracted the attention of many researchers. Luscombe et al. [5] outlined the practice of web scraping and how it works, and compared it to other methods in the social sciences, showing that data scraping helps researchers answer new questions, replaces the limitations of official data, overcomes barriers to access, and reinvigorates the values of sharing, openness, and trust in the social sciences. Ma et al. [6] used bidirectional long short-term memory deep active learning and conditional random field to capture and label the features of different information in the corpus and combined the continuous bag word embedding model and skid-gram word embedding model to capture the text features of Chinese named entity recognition from the unlabeled corpus. The research results show that, compared with other models, the word-embedding self-guided deep active learning method based on unlabeled medical corpora has better performance. Feature selection requires a large amount of prior knowledge because of the length and noise of the data. Luo et al. [7] proposed a bidirectional long short-term memory network based on a convolutional attention mechanism to achieve end-to-end lifetime prediction of rotating machinery. The research results show that this method has good prediction accuracy, which is superior to other methods. Surface Electromyography (EMG) signals are spatially sparse owing to the position of the electrodes on the

hand muscles and are time-dependent owing to activity performance over a period of time. Karnam et al. [8] proposed a hybrid Electromyography-Hand (EMGH) and Net architecture for bidirectional long short-term memory networks based on a convolutional attention mechanism. Encoding the interchannel dependence and time dependence of surface EMG signals for hand movement classification showed accuracies of 95.77%, 95.9%, 91.65%, and 98.33% in the NinaPro DB1, DB2, DB4, and UCI gestures, respectively. This exceeds that of the current state-of-the-art models.

Sentiment analysis helps understand consumer attitudes towards goods. In addition, the construction of multidimensional product feature models helps e-commerce platforms understand users' shopping behavior preferences more deeply. Several scholars and scientists have conducted relevant studies. Aiming at the importance of Internet evaluation in influencing consumers' purchasing habits and improving global communication among consumers, Shiny [9] presented a general classification method for emotional polarity and provided a comprehensive description of the classification process. The results showed that its performance was evaluated by 10x cross-validation. In the field of natural language processing, the execution of sentiment analysis is challenging owing to the insufficient amount of labeled data. Alghamdi et al. [10] proposed a search optimization algorithm with a deep learning function, applied the Glove technology to generate feature vectors, and used a gated loop unit model based on self-head multiple attention to identify and classify emotions. The research results show that this model performs better than most advanced methods. To analyze terrorism-related tweets, Djaballah et al. [11] proposed an improved use of weighted term dictionaries, Word2vec methods, and triples, and categorized them based on fuzzy logic, specifically to analyze terrorism-related sentiment analysis on Twitter. The research results show that the accuracy of the new technology in radial detection is 75–78%, and the accuracy of radiality detection is 61–64%. In view of the problems of sparse data and changing user preferences faced by traditional collaborative filtering algorithms in online commerce, Patil et al. [12] proposed a recommendation system method that combines collaborative filtering and content-based filtering. The research results show that the system provides accurate recommendations for new and old users by analyzing user purchasing behavior and feedback. In view of the limitations of traditional collaborative filtering methods in multi-criteria recommendation systems, such as the shortcomings of single-standard collaborative filtering (CF) in dealing with cold start problems and related-based similarity problems, Singh et al. [13] compared and evaluated traditional collaborative filtering, matrix decomposition and deep matrix decomposition techniques. The results show that the deep matrix decomposition technique performs better than other traditional methods on multi-criteria data sets and can provide more accurate recommendations.

To sum up, in the field of information extraction, although various algorithms have been proposed to improve the efficiency and accuracy of data capture, they still face the challenges of data sparsity and noise interference when processing large-scale and dynamically changing web data. In terms of emotion analysis, although existing studies can classify consumers' emotional tendency, they are still insufficient in multidimensional emotion analysis, emotional trend prediction and in-depth exploration of the reasons behind emotions. In addition, in the field of recommendation system, the traditional collaborative filtering method has limitations in dealing with the cold start problem of new users and multi-standard recommendation, and the existing improved methods have not fully solved these problems. The research aims to fill these gaps by integrating information extraction, sentiment analysis and recommendation system techniques. Through the improved S-PageRank algorithm and the dynamic topic library generation strategy, the accuracy and efficiency of cross-platform information extraction are improved, and the problems of data sparsity and noise interference are solved. Secondly, combining topic model and deep learning technology, a bidirectional short-duration memory network and comprehensive attention mechanism model integrating theme word features are proposed to achieve more accurate multidimensional emotion analysis and emotion trend prediction. Finally, by combining collaborative filtering and content-based filtering methods, an intelligent recommendation system is designed to provide personalized recommendations for new and old users.

The contribution of the research is to propose a web page information extraction method combining the S-PageRank algorithm of keyword weight allocation and the dynamic topic library generation strategy, which effectively improves the

accuracy and accuracy of the topic crawler, and solves the problems of "local optimal solution" and "topic deviation" which are easy to appear in the traditional algorithm in the massive web pages. The introduction of template based web page information automatic extraction strategy, through the semantic annotation of DOM tree and template matching, the rapid and accurate extraction of commodity information is realized, and the information extraction time is greatly shortened.

## 3. Methodology

This study focuses on cross-platform e-commerce information extraction and intelligent product recommendation. First, the Shark-PageRank algorithm and dynamic topic repository technique are used to determine a queue of topic pages to enhance crawler accuracy. Subsequently, a template-based strategy was introduced to extract product information quickly and accurately. The LDA and Bi-LSTM hybrid attention mechanism models were used to extract the commodity subject words, and the review sentiment values were calculated to synthesize the user preferences to achieve intelligent commodity recommendations.

### 3.1. Methods for extracting information from cross-platform e-commerce topic pages

Examples of cross-platform information extraction include grabbing product information from multiple e-commerce platforms (such as Taobao, JD.com, and Amazon) or extracting user comments and feedback from social media platforms (such as Weibo and Douyin). The data on these platforms usually exist in different formats and structures, so the corresponding technical means are required to parse and extract [14,15]. Therefore, this paper introduces an S-PageRank algorithm that relies on keyword weight allocation to crawl topic-related webpages and introduces a template-dependent strategy to extract topic-related information. The S-Search algorithm is a heuristic topic crawler based on web content, which predicts the correlation between each sub-link and the topic to select the product information that best matches the user's query in the massive web pages [16,17]. The relevance of each sublink to the topic is calculated using Eq. (1).

$$
\begin{aligned}
&Potential\_Score(child\_url) \\
&= a \times Neighbour(child\_url) + (1-a) \times Inherited(child\_url), (a < 1)
\end{aligned} \tag{1}
$$

In Eq. (1), *Inherited* indicates that the current node inherits the score of the parent node, as shown in Eq. (2).

$$
\begin{aligned}
&Inherited(child\_url) \\
&= \begin{cases} \beta \times Sim(topic, current\_url), & Sim(topic, current\_url) \geq \mu \\ \beta \times Inherited(topic, current\_url), & Sim(topic, current\_url) \geq \mu \end{cases}
\end{aligned} \tag{2}
$$

In Eq. (2), *current_url* denotes the current node, $Sim(topic, current\_url)$ denotes the similarity between the web content and topic of the current node, and $\mu$ denotes the relevance threshold. The Shark-Search algorithm is based on the Vector Space Model (VSM) to calculate the similarity between texts, which is calculated as shown in Eq. (3) [18,19].

$$
Sim(q, p) = \frac{\sum\limits_{k \in q \cap p}^{n} W_{qk} \times W_{pk}}{\sqrt{\left| \sum\limits_{k \in q}^{n} W_{qk}^2 \right|} \times \sqrt{\left| \sum\limits_{k \in p}^{n} W_{pk}^2 \right|}} \tag{3}
$$

In Eq. (3), $q$ denotes the thesaurus, $p$ denotes the text collection, and $W_{pk}$ denotes the importance of the feature word, $k$, to the topic in the text collection. Because the S-Search algorithm selects less anchor text information and is a local optimal solution algorithm, a "local optimal solution problem" may occur when facing a large number of webpages [20,21]. The

study also proposed the PageRank algorithm, which evaluates the importance of a webpage based on its links. PageRank is determined by the number of incoming links, the PageRank value of the links, and the number of outgoing links of the webpage, which is calculated using Eq. (4).

$$PageRank(A) = \frac{(1-d)}{N} + d \times \sum_{i=1}^{n} \frac{PageRank(T_i)}{C(T_i)}$$

(4)

In Eq. (4), $PageRank(A)$ denotes the $PageRank$ value of page $A$; $PageRank(T_i)$ denotes the $PageRank$ value of page $T_i$; $C(T_i)$ denotes the number of links, and $d$ denotes the damping coefficient. The PageRank algorithm evaluates the importance of a page and guides web crawlers to crawl it, but there are problems such as "topic drift." Therefore, we propose a Shake-PageRank positioning method based on keyword weighting. It combines the Shake-Search and PageRank algorithms to describe topics accurately [22,23]. To find the user shopping page, Term Frequency-Inverse Document Frequency (TF-IDF) is used to extract feature words to build a thesaurus, calculate the similarity between keywords and feature words, and update the weights of the high-similarity words, which are based on the word frequency and inverse document frequency, as shown in Eq. (5).

$$w_{i,j} = tf_{i,j} \times idf_{i,j} = \frac{n_{i,j}}{\sum_{k} n_{k,j}} \times \log \frac{|D|}{|j : t_i \in d_j|}$$

(5)

In Eq. (5), $w_{i,j}$ denotes the feature item weights, $tf_{i,j}$ denotes word frequency, $n_{k,j}$ denotes frequency, and $idf_{i,j}$ denotes inverse document frequency. The weights were normalized and calculated, as shown in Eq. (6).

$$w_{i,j} = \frac{tf_{i,j} \times \log(\frac{D}{|\{j:t_i \in d_j\}|} + 0.01)}{\sqrt{\sum_{i=1}^{n} \left[ tf_{i,f} \times (\frac{D}{|\{j:t_i \in d_j\}|} + 0.01) \right]^2}}$$

(6)

Next, the study constructs high-dimensional word vectors for each feature word and calculates the semantic similarity between the feature word and topic word using cosine similarity, as shown in Eq. (7).

$$\cos(W_i, W_j) = \frac{\sum_{k=1}^{N} (W_i^k \times W_j^k)}{\sqrt{\sum_{k=1}^{N} (W_i^k)^2} \times \sqrt{\sum_{k=1}^{N} (W_j^k)^2}}$$

(7)

In Eq. (7), $W_i^k$ denotes the dimension of the word vector of word $W_i$. After analyzing the advantages and disadvantages of Shark-Search and PageRank, the study combines the two to create the Shark-PageRank algorithm, which scores web pages for topical relevance in terms of text and link relationships and crawls web pages based on the scores. This alleviates the "local optimality" problem of Shark-Search and the "topic drift" problem of PageRank [24,25]. The calculation formula is shown in Eq. (8).

$$
\begin{aligned}
PageRank\_Score(child\_url) = \\
A \times Inherited(child\_url) + B \times Neighbor(child\_url) \\
+C \times PageRank(child\_url)
\end{aligned}
$$

(8)

In Eq. (8), $A, B, C$ is in the range of 0–1 and $A + B + C = 1$. The Shark-PageRank algorithm is used to calculate the webpage topic relevance score, and links to webpages whose score exceeds the threshold $\beta$ are added to the topic webpage queue. Threshold $\beta$ was calculated using Eq. (9).

$$\beta = \frac{\sum_{i=1}^{n} url_i}{n}$$

(9)

In Eq. (9), $url_i$ denotes the topic relevance score of link. Webpage data are mostly in HTML tags, which makes it difficult to extract structured data using conventional methods. The data display pattern can be located by transforming HTML into a tree structure through a Document Object Model (DOM). This paper proposes a webpage information extraction method for templates based on this. First, DOM is used to remove the noise and locate the theme information, summarize the path to make the extraction template, and then extract the information through template matching, as shown in Fig 1 [26,27].

As shown in Fig 1, the HTML document is first preprocessed and converted into a DOM tree. Considering the aggregation characteristics of merchandise information, a domain semantic dictionary was used to annotate the leaf nodes of the DOM tree, compute their semantic values, pinpoint the nodes where the information is concentrated, and exclude noise [28]. The merchandise attribute information is extracted from these nodes, and an extraction template consisting of Xpath paths and regularity is summarized for similar pages. The multidimensional semantic dictionary is used for semantic

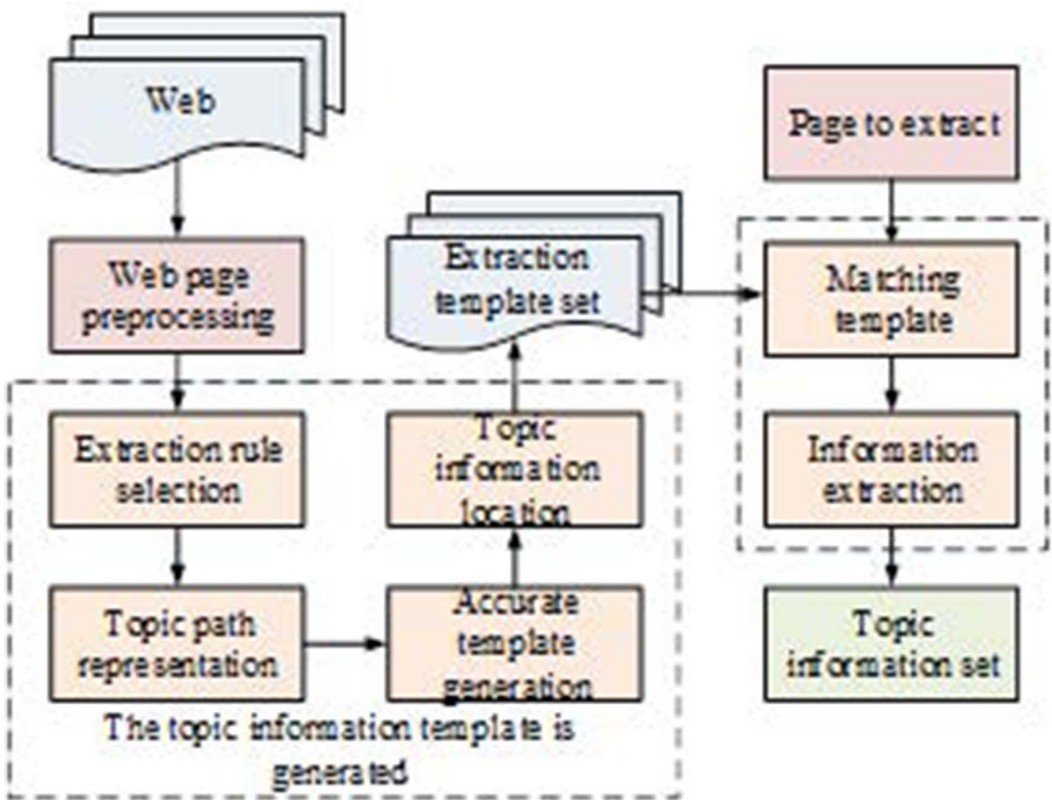

**Fig 1. The basic process framework of topic information extraction.**

annotation of the DOM tree, which is categorized by product attributes, e.g., "product review" and "product name" are different attributes, while "price" and "promotional price" are different attributes. Price" and "Promotion Price" had the same attributes. Fig 2 shows a sample DOM tree after annotation, $N$ represents the node of "product name" information; $M$ represents the node of "delivery mode" information; $P$ represents the node of "price" information; $C$ represents the node of "number of reviews" information; represents the node of "number of reviews" information; represents the node of "price" information; represents the node of "number of reviews" information. $U$ is the node of "number of comments, " and is the node with no joints.

### 3.2. Commodity recommendation based on sentiment analysis of online shopping reviews

To gain a deeper understanding of users' attitudes and feelings towards products, the study needs to further analyze the reviews left by users. Therefore, this study focuses on intelligent product recommendations using a sentiment analysis of online shopping reviews. The study uses the Latent Dirichlet Allocation (LDA) model to extract review topic words, and semantically similar words are grouped into the same topic. LDA is a widely used probabilistic topic model, which is regarded as the clustering of topic words. The idea is to select topics in documents, and then select words from topics, forming a "document-topic-topic-word" distribution [29,30]. The structure of the LDA topic model is illustrated in Fig 3. $M$ is the number of documents, $K$ is the number of topics, $N_m$ is the number of words in the $m$ th document, $a$ and $\beta$ are the prior distribution parameters of the topics and words of the topics in the documents, $\vartheta_m$ represents the topic distribution of the $m$ th article, $\emptyset_k$ is the distribution of the words of the $k$ th topic, and $Z_{m,n}$ and $W_{m,n}$ refers to the $n$ th word of the $m$ th article.

When extracting commodity subject terms using the LDA model, the value of the number of topics in $K$ needs to be determined. It is selected by calculating the perplexity of each $K$ value, which is a criterion for assessing the predictive effectiveness of probabilistic models. A low perplexity means that the model predicts more accurately (see Eq. (10), to obtain an exact calculation.

$$Perplexity(D) = \exp(-\frac{\sum \log p(w)}{\sum_{d=1}^{M} N_d})$$

(10)

In Eq. (10), *Perplexity* denotes the degree of confusion and $p(w)$ denotes the probability of word occurrence in the test set. As mentioned above, the study designed the overall framework of the model, as shown in Fig 4, which

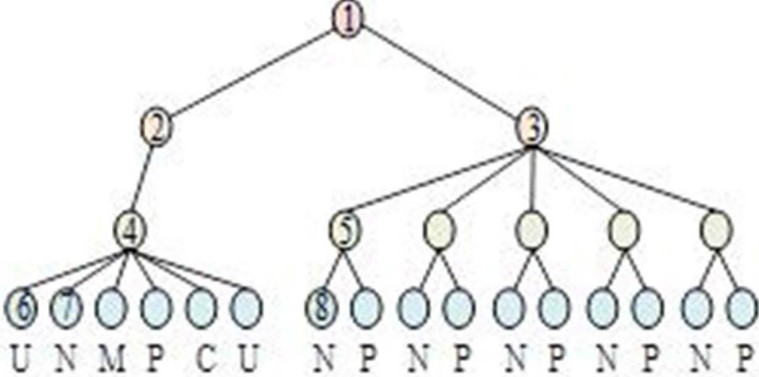

**Fig 2.  DOM tree example after semantic annotation.**

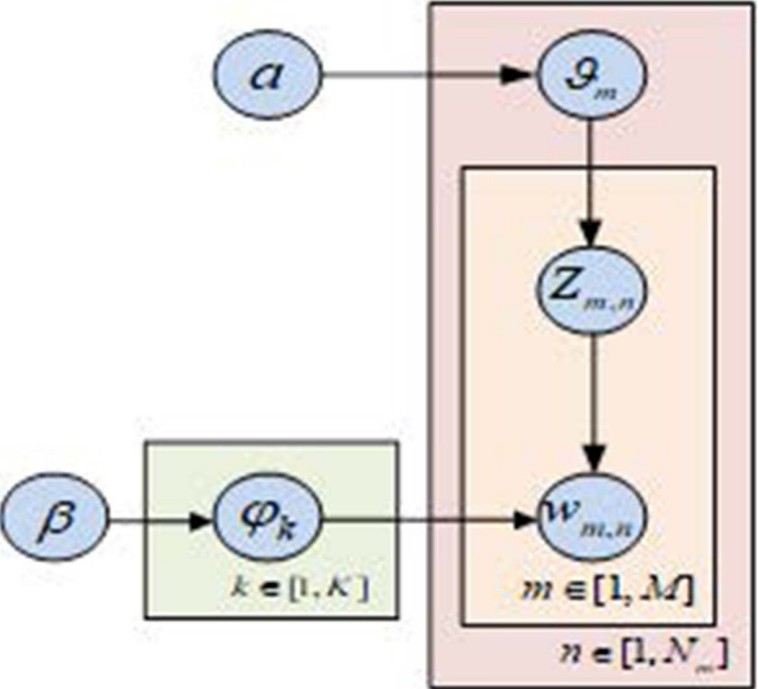

**Fig 3. LDA motif configuration diagram.**

consists of five parts: input, memory, attention, fully connected linear transformation, and the Softmax classifier. The input module transforms comment text and subject words into a sequence of word vectors. The memory module, which often uses LSTM and its variants, constructs sentence-based topic word-specific memory matrices. The attention module captures key information related to the sentiment tendencies of subject words. The fully connected linear transform module linearly transforms the output of the attention module to provide easily processable vectors for the prediction. The Softmax classifier then predicts the sentiment tendency of the comment to the topic word based on this output [31,32].

The Bi-LSTM+Attention Mechanism model incorporating topic word features is illustrated in Fig 5. The structure of the model starts from the input layer, passes through the vector layer, Bi-LSTM layer, attention mechanism layer, and then to the fully connected layer and softmax layer, and finally outputs the probability of the comment text for each sentiment tendency [33–35].

For topic words containing multiple words, the average of the word vectors was taken as the topic word vector, which was calculated using Eq. (11).

$$w^t = \frac{1}{m} \times \sum_{i=1}^{m} w_i^t$$

(11)

This study employs Bi-LSTM to extract the vectorial features of comment text and topics. Bi-LSTM combines forward and backward LSTM layers to capture contextual information of the text. It uses the combined sequence of comment text vectors and topic vectors $w = [w^s, w^t]$ as input, and the computational procedure is referred to in Eq. (12).

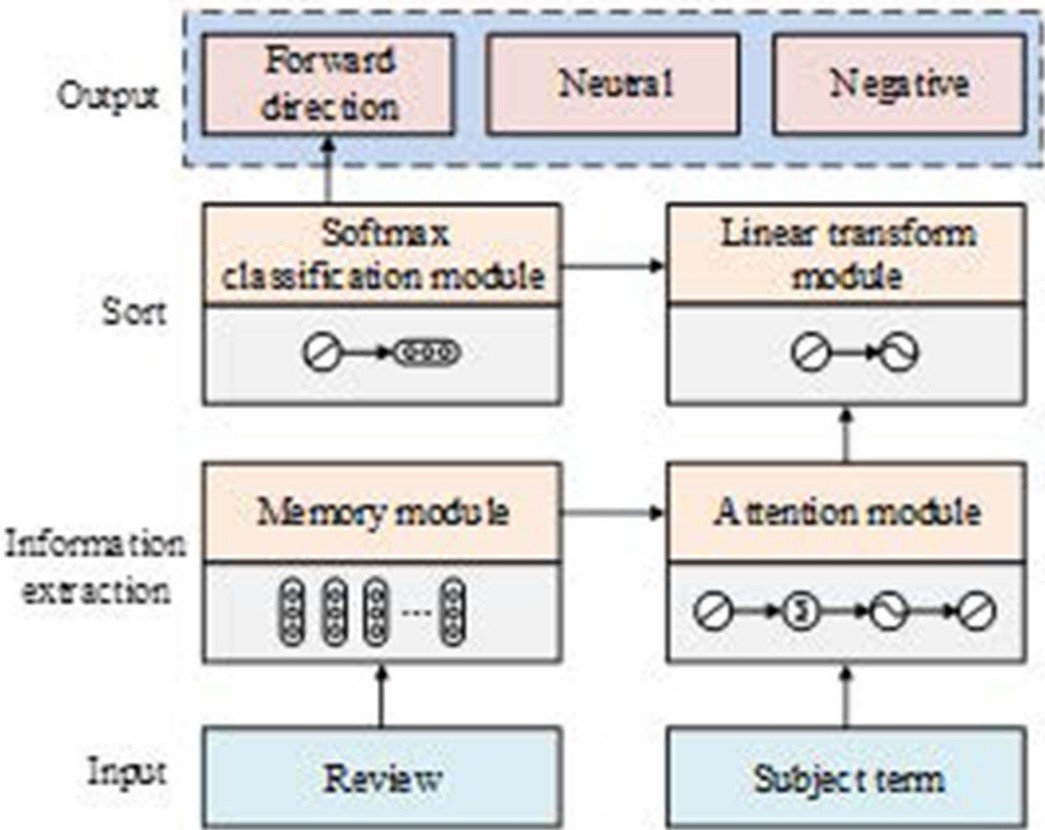

**Fig 4. The whole frame diagram of the mould.**

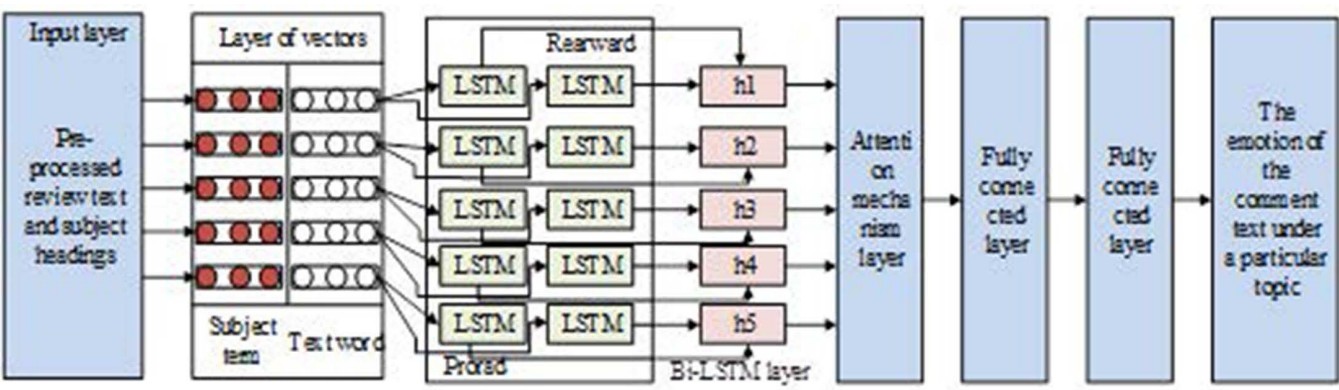

**Fig 5. Structure diagram of Bi-LSTM+ attention mechanism model integrating theme word features.**

$$h_t^{forward} = LSTM^{forward}(h_{t-1}, w, c_{t-1})$$
$$h_t^{backward} = LSTM^{backward}(h_{t-1}, w, c_{t-1})$$
$$h_t = [h_t^{forward}, h_t^{backward}]$$

(12)

Eq. (12) shows the formula for LSTM. $w$ is the input vector of the moment $t$; $c_{t-1}$ and $h_{t-1}$ are the memory and hidden states of the moment $t-1$; $c_t$ and $h_t$ are the memory state and output of the moment $t$, respectively; and the output sequence of the entire Bi-LSTM is $H = \{h_1, \cdots, h_j, \cdots, h_n\}$. The attention mechanism layer receives the output vector $H \in R^{d \times n}$ and the subject word vector $w_i^s \in R^{d \times l}$ of Bi-LSTM, and the output vector $s \in R^{d \times l}$ is the weighted sum of $H$, which is calculated by referring to Eq. (13).

$$s = \sum_{i=1}^{n} a_i h_i$$

(13)

In Eq. (13), $a_i$ denotes the weight of $h_i$. The output of the Attention Mechanism layer is a dimensional vector, where dim is the preset hyperparameter. The model ultimately has four output categories; therefore, the Softmax layer outputs a four-dimensional vector representing the probability of each category. To connect these two layers, a fully connected layer is inserted, which implements a linear transformation from dim-dimensional vectors to four-dimensional vectors, as detailed in Eq. (14).

$$Output_i = w_l \times m_i + b_l$$

(14)

In Eq. (14), $w_l$ denotes the weight matrix and $b_l$ denotes the bias. The softmax layer, which normalizes each category and outputs its probability, is used for multi-classification problems, as shown in Eq. (15).

$$a_i = \frac{\exp(Output_i)}{\sum_{i=1}^{c} \exp(Output_i)}$$

(15)

In Eq. (15), $c$ denotes the number of categories and $a_i$ denotes the probability corresponding to the category. To consider the contribution of the position and part of the word in the word sequence to topic sentiment, this study introduces a hybrid attention mechanism in combination with the traditional attention mechanism. First, a weighted sum is made of the importance of the topic word in the preceding, following, and full text. Next, positional weights were added as the final expression. Finally, the softmax function was used to derive the sentiment tendency [36–38]. Fig 6 shows a model structure diagram of the hybrid attention mechanism.

In Fig 6, the feature vector $\{h_1, h_2, \cdots, h_n\}$ consists of the vector $\{w_1, w_2, \cdots, w_n\}$ and sentiment features extracted by Bi-LSTM, where $w^t$ represents the subject word vector. In the hybrid attention layer, the feature vector first extracts sentiment features through three attention mechanisms: the global, left, and right sides of the topic word. The result of each mechanism is then multiplied by the positional weight of the word. Finally, the weighted sum of the three results was the output of the hybrid attention layer.

In order to further explore the emotional trend and the reasons behind the emotion in the e-commerce recommendation system, the study introduces the affective trend prediction. The time series analysis method is used to model the sentiment value of user comments. First, the user comments were sorted chronologically and the sentiment score for each point in time was calculated. Then, sliding window technology is used to smooth the emotion score to reduce the impact of short-term fluctuations. Then, autoregressive moving average (ARMA) model or long short-term memory network (LSTM) were used to model the time series of emotion scores. By training the model, it is possible to predict emotional trends at

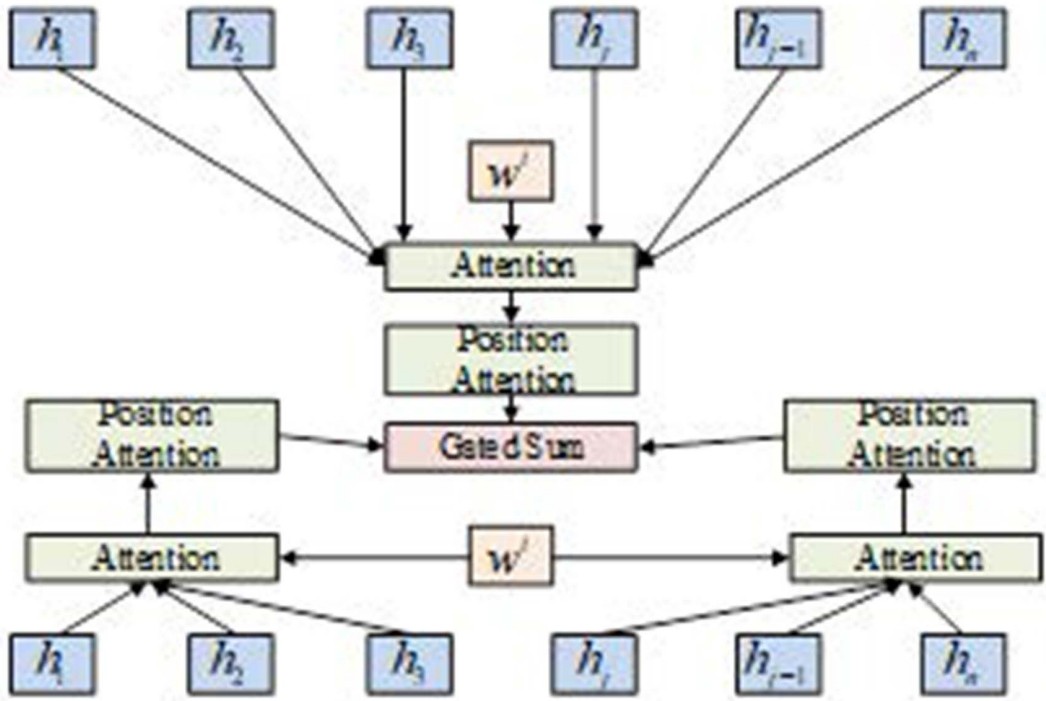

**Fig 6. Mixed attention mechanism layers.**

future points in time, thus providing early warning for businesses to adjust product strategies or marketing programs in a timely manner. The purpose of the analysis of the reasons behind emotion is to explore the potential factors of the user's emotional tendency. Therefore, the method of causality analysis is adopted. First, text mining techniques are used to extract emotion-related keywords and phrases from user reviews that may be related to specific attributes of the product (such as price, quality, features, etc.). Then, the association rule mining algorithm is used to analyze the association relationship between these keywords and phrases to identify the factors that may cause the emotion change.

## 4. Experimental validation and analysis of theme-based information extraction and sentiment analysis for web pages

This experiment first provides an in-depth analysis of topic-based and template-based information extraction from target webpages and compares the effectiveness of both methods. Then, the LDA method was used to extract information from topic words and to analyze the sentiment of user comments under a specific topic. Finally, a recommendation model was designed and validated in conjunction with sentiment calculation to provide users with more accurate content recommendations.

### 4.1. Theme-based target webpage information extraction results and analysis

In making recommendations for e-commerce users, the research mainly uses the following types of data. The first is user review data, through the analysis of users' comments on goods, in-depth understanding of users' satisfaction, preferences and potential demand for goods. The second is the user behavior data, which includes: purchase history, the user's purchase record reflects its actual purchase behavior and preferences. Browsing behavior: The user's browsing history (such as clicks, time spent, etc.) reveals the user's level of interest in certain products. Search history: Users' search keywords

 

can reflect their current needs and interests. The third is the commodity attribute data, including the detailed description, function, characteristics and other information of the commodity, which can help the recommendation system to better understand the characteristics of the commodity, so as to achieve content-based recommendation. It also includes product classification information (such as electronics, clothing, household goods, etc.), which is used to build a similarity relationship between products and help the recommendation system find products that are similar to the user's historical behavior. Price information: Product price ranges and promotions are used to meet users' budget needs and provide more cost-effective recommendations. The fourth is user profile data, including demographic information, interest labels, and social network data. User reviews play an important role in e-commerce recommendation function, which can not only improve the accuracy and personalization of recommendations, but also enhance user engagement and user experience of the platform.

This experiment uses the Jingdong mobile phone catalogue as the initial URL and dynamic topic vector input and crawls 5000 web pages in Jingdong using four algorithms: Shark-Search, PageRank, Shark-PageRank, and keyword-weighted Shake-PageRank combined with the Hertrix framework. Because of the large number of web pages on the Internet, the experiments did not use the check-all rate evaluation but only the check-accuracy evaluation. A comparison of the precisions of the four methods is shown in Fig 7.

In Fig 7, the accuracy rate of most algorithms increased, but the performance of PageRank algorithm decreased because it only considered the link structure and ignored the relevance of the page to the topic, resulting in "topic deviation." The algorithm in this study further introduces a dynamically updated thesaurus, and the experimental results show that its accuracy is higher than that of S-PageRank.

## 4.2. Template-based target webpage information extraction results and analysis

The experiment crawled 5,000 mobile phone sales pages on two e-commerce platforms, Jingdong and Suning.com. The selected product attributes were name, number, price, promotion, delivery method, number of favorites, number of

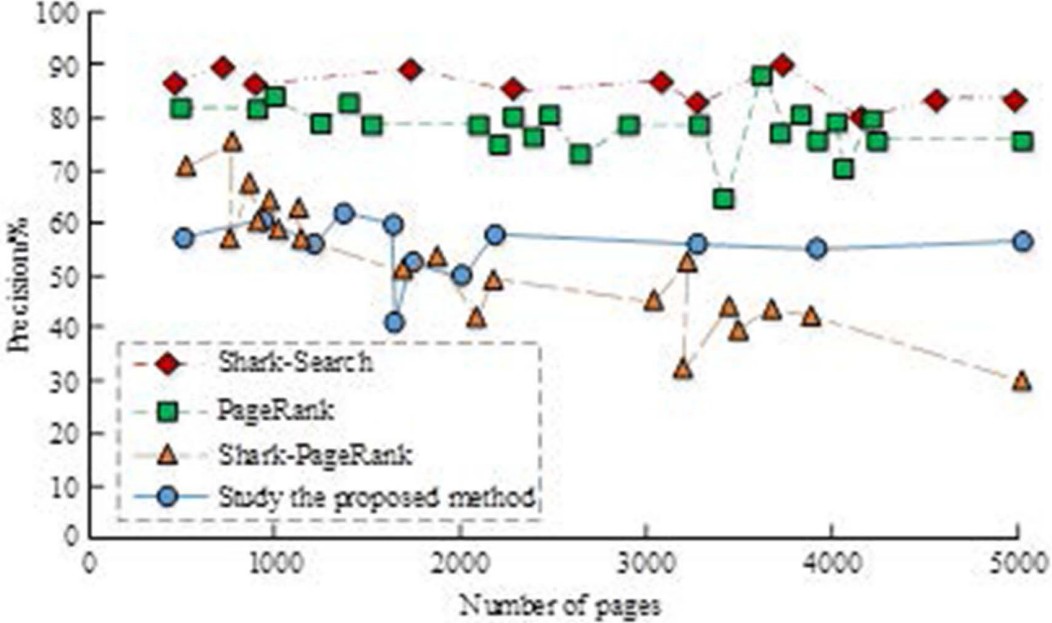

**Fig 7. Comparison of precision of four thematic crawler algorithms.**

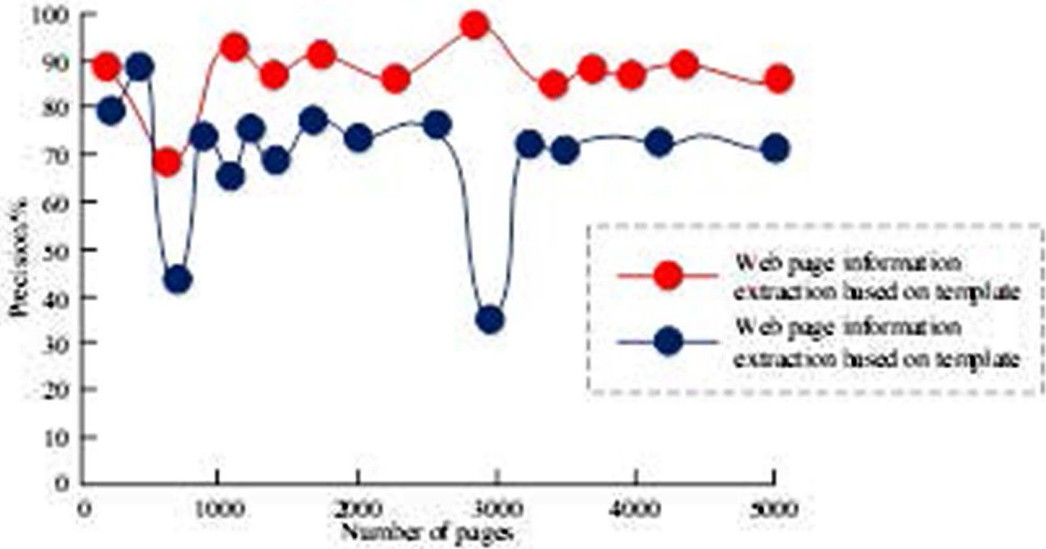

**Fig 8. Comparison of accuracy of two web information extraction methods.**

comments, and various ratings. Information was extracted from these pages using the DOM tree and template-based methods. A comparison of the two algorithms in terms of checking accuracy is shown in Fig 8.

Fig 8 shows the accuracy of the two information extraction methods, DOM tree-based and template-based. The DOM-based method is affected by noise and has a lower checking accuracy, averaging at 80%, whereas the template-based method achieves a higher checking accuracy, averaging at 90%, by removing noise. To verify that the template method could shorten the extraction time, a controlled experiment was conducted, and the results are shown in Table 1.

As can be seen from Table 1, both the structural semantic quotient-based approach and the template-based approach perform well in extracting topic information, and both have check accuracy rates of approximately 90%. Although both had similar check accuracy rates, there was a significant difference in the processing time. The template-based approach is significantly faster than the structural semantic quotient-based approach because it uses templates and avoids parsing a page every time.

### 4.3. LDA-based topic word extraction with results and analysis

In this experiment, the LDA topic model was used to investigate perplexity at different values of $K$. This study aimed to determine the optimal number of topics by observing changes in perplexity. Fig 9 shows the perplexity degree of the LDA topic model under different $K$ values.

The LDA topic model is used in Fig 9 to explore the effects of different numbers of topics on the model effectiveness. It can be observed that the model performs best when the $K$ value is set to 7. This is because, at this $K$ value, the LDA model's perplexity reaches its lowest point, which in turn means that it has the highest prediction accuracy for this number of topics.

**Table 1. The effect of template on topic information extraction.**

| Template | Platform | Page count | Correct extraction number | Error extraction | Precision ratio/% | Time/s |
|---|---|---|---|---|---|---|
| Yes | Jingdong | 1000 | 887 | 113 | 89 | 369 |
| | Suning | 1000 | 909 | 91 | 90 | 307 |
| No | Jingdong | 1000 | 923 | 77 | 92 | 10971 |
| | Suning | 1000 | 911 | 89 | 91 | 9868 |

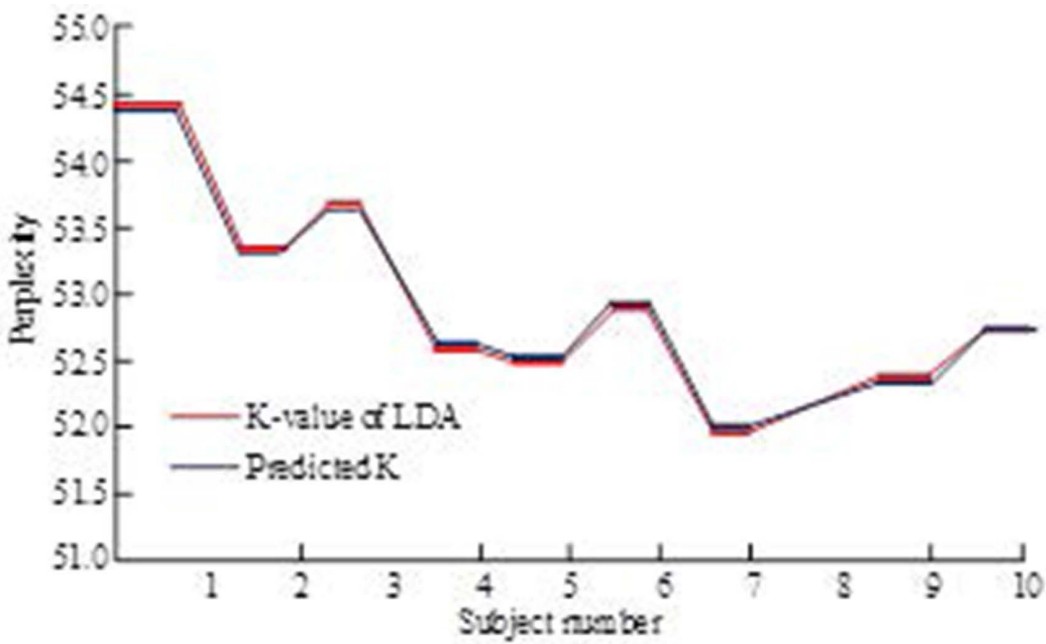

**Fig 9. Confusion degree of LDA subject model under different K values.**

### 4.4. Sentiment analysis of user reviews and recommendation models for specific topics

To achieve this goal, a Bi-LSTM + Hybrid Attention Mechanism model incorporating topic word features was chosen. To further validate the practical effectiveness of Bi-LSTM and the hybrid attention mechanism in sentiment analysis, we designed a set of controlled experiments, as shown in Fig 10.

As shown in Fig 10, the ATAE-LSTM model outperformed both the TD-LSTM and TC-LSTM models in terms of evaluation metrics, which suggests that the attention mechanism effectively captures important contextual information. The performance of the ATAE-LSTM model, when Bi-LSTM + Attention was introduced, further improved in terms of average accuracy, recall, and Macro-F1, which reveals the stronger ability of bi-directional LSTM to capture long-term dependencies relative to unidirectional LSTM. Next, an in-depth sentiment analysis was conducted on 523 user reviews of three mobile phones. This analysis aimed to understand how consumers feel about the specific features or functions of each mobile phone. As shown in Fig 11, the sentiment rating values of the three mobile phones for different topics, such as battery, camera, design, performance, and price, can be clearly observed.

According to Fig 11, the three mobile phones Phone A, Phone B, and Phone C exhibit some differences in the sentiment ratings of different themes. Specifically, Phone A obtains the highest sentiment value on the themes of "Battery" and "Design", which are 0.85 and 0.9 respectively, while the sentiment value on the theme of "Performance" is 0.7. "Phone B has a sentiment value of 0.76 for "Camera" and 0.9 for "Price, "indicating that it receives the highest rating for price. The rating of Phone C is slightly ahead on the "Camera" theme with a sentiment value of 0.79, but slightly behind on the "Performance" theme with a sentiment value of 0.68.

### 4.5. Effectiveness analysis of intelligent recommendation of web information based on improved S-PageRank algorithm

The performance of the improved S-PageRank algorithm (Algorithm 1) is further analyzed and compared, including the C4.5 decision tree algorithm (Algorithm 2), classification algorithm based on Bayes' theorem (Algorithm 3), AdaBoost

 

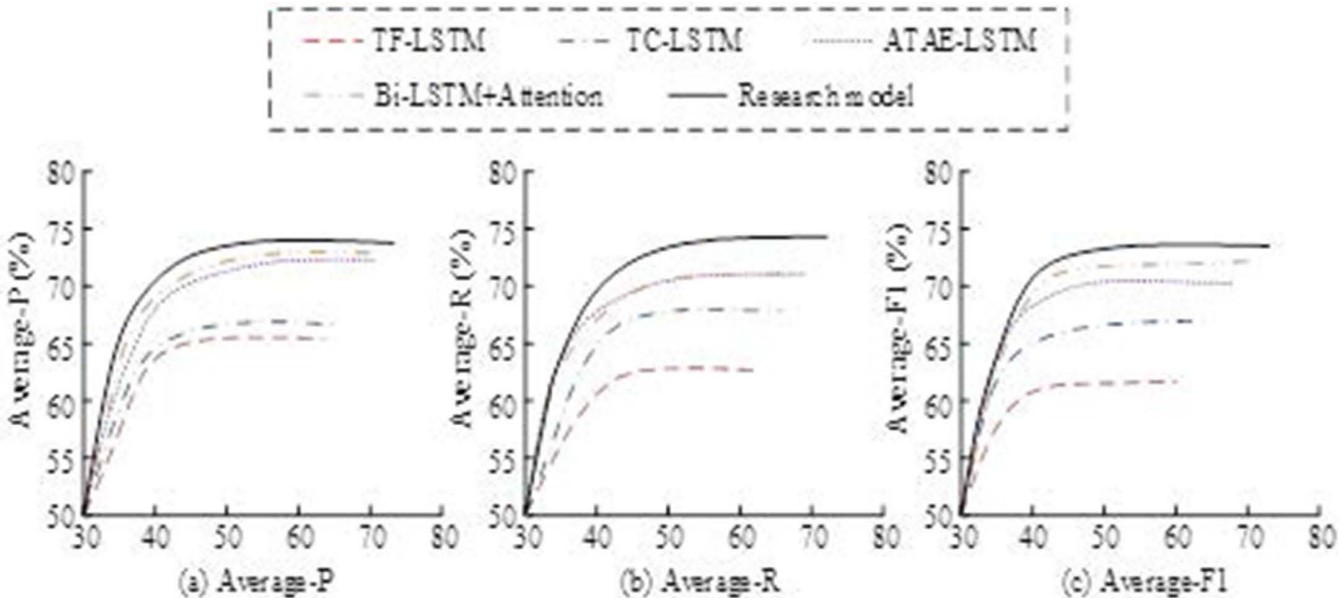

Fig 10. The fruit ratio of the controlled experiment.

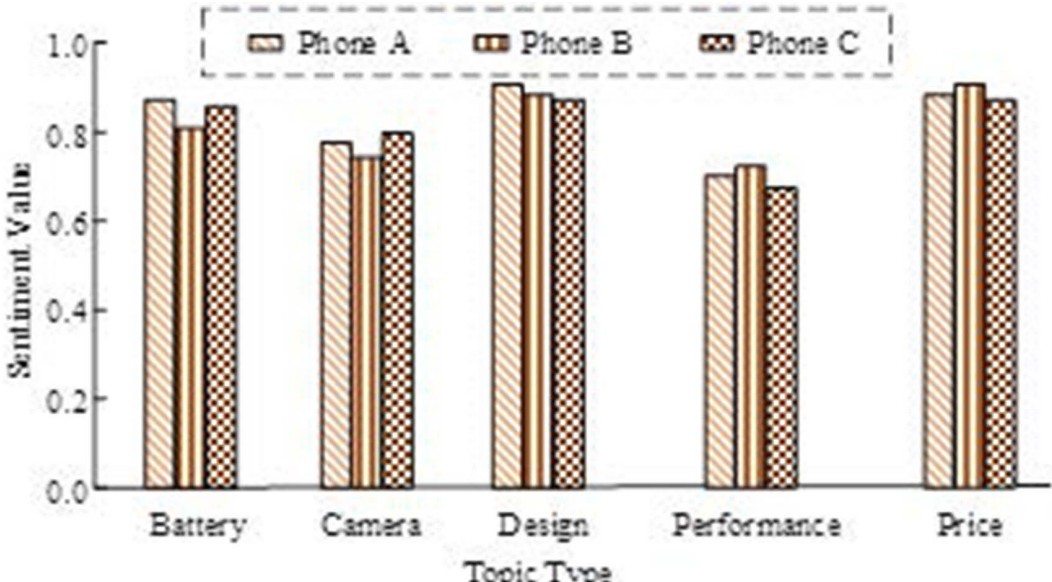

Fig 11. Emotional analysis of three mobile phones in different themes.

integrated learning algorithm (Algorithm 4), and PageRank page sorting algorithm (Algorithm 5). The accuracy and recall rates of the five information extraction algorithms are shown in Fig 12.

In Fig 12a, the accuracy of Algorithm 1 is the highest, exceeding 90%, indicating that Algorithm 1 has the best performance. In Fig 12b, the recall rate represents the proportion of positive samples correctly identified by the algorithm for all

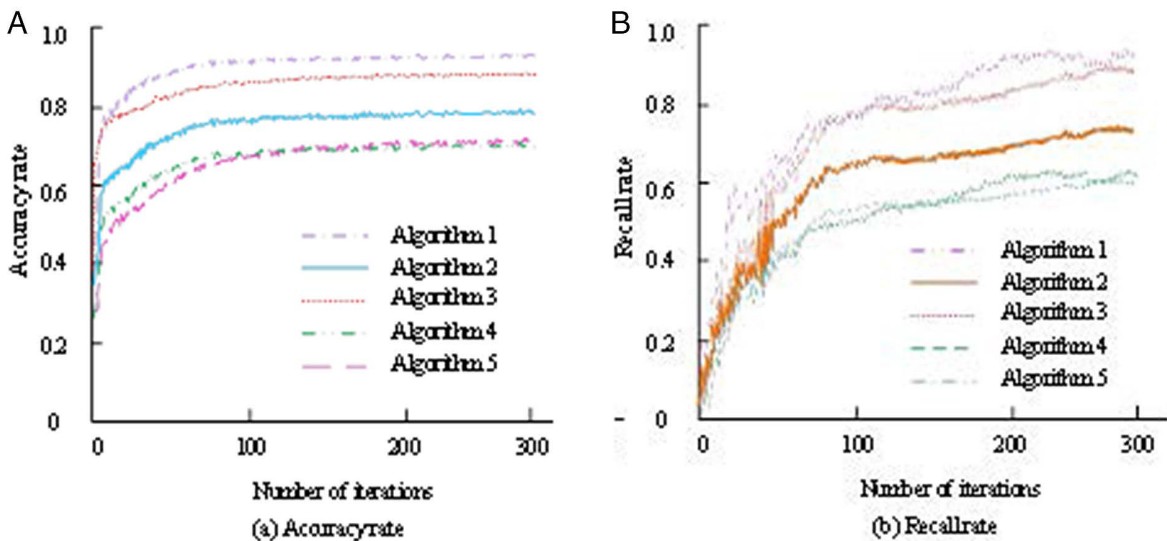

**Fig 12. Accuracy and recall rate of five algorithms.**

the actual positive samples. Algorithm 1 has the highest recall rate, which eventually exceeds 90%, indicating that Algorithm 1 has the strongest ability to identify positive samples.

In order to verify the performance advantages of the e-commerce recommendation system proposed in this paper in multidimensional emotion analysis, especially the ability to predict emotional trends and analyze the reasons behind emotions, two sets of experiments were designed and compared with the current most advanced e-commerce recommendation system (the system used by Amazon ([12] Patil P et al. (2024)) and Netflix([13] Singh R et al. (2024))), as shown in Table 2.

In Table 2, the MSE of the proposed system is 0.035, which is much lower than Amazon's 0.052 and Netflix's 0.048, indicating that this system has obvious advantages in the accuracy of emotional trend prediction. At the same time, the MAE of this system is 0.021, which is lower than Amazon's 0.033 and Netflix's 0.030, which further proves its superiority in forecasting accuracy. In addition, the system's prediction accuracy reached 92.5%, higher than Amazon's 88.0% and Netflix's 89.5%, which indicates that the system can more accurately predict the changing trend of user emotion. The association rule confidence of the system is 0.85, which is higher than Amazon's 0.78 and Netflix's 0.80, indicating that the system can more accurately identify the key factors behind the emotion. At the same time, the association rule support of this system is 0.60, which is also higher than Amazon's 0.52 and Netflix's 0.55, indicating that this system has advantages

**Table 2. Performance comparison of multidimensional sentiment analysis of e-commerce recommendation systems.**

| Experimental index | Our system | Amazon recommendation system | Netflix recommendation system |
| --- | --- | --- | --- |
| Mean square error (MSE) | 0.035 | 0.052 | 0.048 |
| Mean absolute error (MAE) | 0.021 | 0.033 | 0.030 |
| Prediction accuracy | 92.5% | 88.0% | 89.5% |
| Model training time (seconds) | 120 | 180 | 150 |
| Association rule confidence | 0.85 | 0.78 | 0.80 |
| Association rule support | 0.60 | 0.52 | 0.55 |
| Affective cause analysis accuracy | 90.0% | 85.0% | 87.0% |
| Analysis time (seconds) | 90 | 120 | 110 |

in the universality and reliability of emotional reason analysis. In addition, the accuracy rate of emotional reason analysis of this system reaches 90.0%, which is higher than that of Amazon (85.0%) and Netflix (87.0%), which further proves its superiority in emotional reason analysis. Although the analysis time of this system is slightly shorter than that of Amazon and Netflix, its significant improvement in analysis accuracy and reliability indicates that this system has higher practicality and accuracy in analyzing the reasons behind emotions.

## 5. Discussion

The research has achieved remarkable results in cross-platform e-commerce web page information extraction and sentiment analysis, especially excelling in precision rate, the accuracy of sentiment analysis and the overall performance of the recommendation system. The S-PageRank algorithm based on keyword weight allocation and the dynamic topic library generation strategy proposed in the research have effectively improved the precision rate of topic crawlers. The precision rate of this method has reached 90%, which is significantly better than the traditional S-PageRank algorithm. Compared with the research of Li et al., this study has significantly improved in both the accuracy and efficiency of information extraction. Although Luscombe et al. emphasized the importance of web scraping, when dealing with large-scale dynamic web page data, their methods still face the challenges of data sparsity and noise interference [39]. The research effectively solved these problems by optimizing the algorithm and introducing the template strategy. When the number of topics of the LDA model is 7, the prediction accuracy of the model reaches its peak, and it is superior to the previous methods in terms of accuracy, recall rate and F-score. The research introduces a comprehensive attention mechanism, taking into account not only the word positions in the comment texts but also the contribution degrees of different parts to the sentiment tendencies, thereby calculating the sentiment values of user comments under different topics more accurately. The MSE of the e-commerce recommendation system proposed in the study is 0.035, the MAE is 0.021, and the prediction accuracy rate reaches 92.5%, all of which are superior to the existing recommendation systems such as Amazon and Netflix. Furthermore, the confidence level and support level of the association rules of the system are also higher than those of Amazon and Netflix, indicating that it has significant advantages in the universality and reliability of sentiment cause analysis. Compared with the recommendation system method of integrated collaborative filtering and content-based filtering proposed by Ding et al., the recommendation system in this study not only performs well in dealing with the cold start problem of new users, but also provides more accurate personalized services in multi-criterion recommendation. Although Patil et al.'s [12] method provides precise recommendations for users by analyzing their purchasing behaviors and feedback, it still faces challenges when dealing with large-scale dynamic data [40]. The research fills these gaps by comprehensively applying information extraction, sentiment analysis and recommendation system technologies.

## 6. Conclusion

To address the challenges of cross-platform product information extraction and intelligent product recommendations, two core strategies were adopted in this study. First, a keyword-weighted Shark-PageRank algorithm and a dynamic topic library strategy were combined in the information extraction session. Second, template-based sentiment analysis was introduced to accurately and rapidly extract commodity information and lay the foundation for commodity recommendations. The experimental data demonstrate that the strategy proposed in this study is significantly better than the Shark-PageRank algorithm in terms of checking accuracy. In terms of information extraction, the template-based approach and DOM tree-based approach have 90% and 80% checking accuracy, respectively, and the template-based strategy has 10% higher checking accuracy. Consumer evaluations of Phones A, B, and C for different topics showed differences. The accuracy of the improved algorithm was the highest, reaching more than 90%, and the recall rate was the highest, reaching 90%. The system in this paper performs well in the two key tasks of emotion trend prediction and emotion reason analysis, and its prediction accuracy and analysis accuracy are better than the advanced recommendation system used by Amazon and Netflix. This indicates that the e-commerce recommendation system proposed in this paper

has significant advantages in multidimensional emotion analysis, which can provide users with more accurate and personalized recommendation services, and also provide merchants with more in-depth user emotion insight. It can be found that the recommender system proposed in this study shows significant advantages in terms of checking accuracy, extraction time, and other evaluation metrics. However, this research has some limitations, and the fine-grained classification of affective computing is still limited. Future work will focus on further optimizing the recommendation algorithm to improve the accuracy and personalization of recommendations by introducing more fine-grained sentiment analysis and multi-dimensional user behavior modeling. In addition, the research will explore how to better integrate real-time user feedback and social network data to enhance the dynamic adaptability and user interaction of recommendation systems.

## Author contributions

**Writing – original draft:** Jinfeng Feng.

**Writing – review & editing:** Jinfeng Feng.

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
