## [Decision Letter · Decision Letter 0]

PONE-D-25-01279E-commerce Recommender System Design Based on Web Information Extraction and Sentiment AnalysisPLOS ONE

Dear Dr. Feng,

Thank you for submitting your manuscript to PLOS ONE. After careful consideration, we feel that it has merit but does not fully meet PLOS ONE’s publication criteria as it currently stands. Therefore, we invite you to submit a revised version of the manuscript that addresses the points raised during the review process. **The authors are advised to revise the manuscript as per the comments raised by reviewers. ** 

We look forward to receiving your revised manuscript.

Kind regards,

Mudassir Khan, Ph.D

Academic Editor

PLOS ONE

**Journal Requirements:**

3. In the online submission form, you indicated that The data that support the findings of this study are available on request from the corresponding author, [initials]. The data are not publicly available due to their containing information that could compromise the privacy of research participants.

**Additional Editor Comments:**

The authors are advised to revise the manuscript as per the comments raised by reviewers.

Reviewers' comments:

Reviewer's Responses to Questions

**Comments to the Author**

1. Is the manuscript technically sound, and do the data support the conclusions?

Reviewer #1: Partly

Reviewer #2: Yes

2. Has the statistical analysis been performed appropriately and rigorously? 

Reviewer #1: No

Reviewer #2: Yes

3. Have the authors made all data underlying the findings in their manuscript fully available?

Reviewer #1: Yes

Reviewer #2: Yes

4. Is the manuscript presented in an intelligible fashion and written in standard English?

Reviewer #1: Yes

Reviewer #2: Yes

5. Review Comments to the Author

**Reviewer #1: ** 1.The paper mentions multi-dimensional emotion analysis and its potential for exploring emotion trend prediction and reason analysis in the introduction, but it does not provide a detailed explanation or exploration of these aspects in the methodology, experiments, or discussion.

2. The paper tackles the idea of ecommerce recommendation system but doesn't provide any benchmark against state-of-the-art e-commerce recommender systems like those used by Amazon or Netflix.

**Reviewer #2: ** Your manuscript provides a broad exploration of “E-commerce Recommender System Design Based on Web Information Extraction and Sentiment Analysis". While the paper addresses significant aspects of web information extraction and sentiment analysis for E-commerce recommender system, some areas require revision to enhance clarity, coherence, and depth of analysis.

6. PLOS authors have the option to publish the peer review history of their article (what does this mean? ). If published, this will include your full peer review and any attached files.

**Do you want your identity to be public for this peer review?** For information about this choice, including consent withdrawal, please see our Privacy Policy .

Reviewer #1: **Yes: ** Md Abdullah Al Kafi

Reviewer #2: No

---

## [Author Response · Author response to Decision Letter 1]

27 Feb 2025

Review Comments to the Author

Reviewer #1:

1.The paper mentions multi-dimensional emotion analysis and its potential for exploring emotion trend prediction and reason analysis in the introduction, but it does not provide a detailed explanation or exploration of these aspects in the methodology, experiments, or discussion.

Reply: Thank you for your question, the aspects of multidimensional emotion analysis have been explained and explored in detail in methodology, experiments and discussions. Related changes are as follows:

In order to further explore the emotional trend and the reasons behind the emotion in the e-commerce recommendation system, the study introduces the affective trend prediction. The time series analysis method is used to model the sentiment value of user comments. First, the user comments were sorted chronologically and the sentiment score for each point in time was calculated. Then, sliding window technology is used to smooth the emotion score to reduce the impact of short-term fluctuations. Then, autoregressive moving average (ARMA) model or long short-term memory network (LSTM) were used to model the time series of emotion scores. By training the model, it is possible to predict emotional trends at future points in time, thus providing early warning for businesses to adjust product strategies or marketing programs in a timely manner. The purpose of the analysis of the reasons behind emotion is to explore the potential factors of the user's emotional tendency. Therefore, the method of causality analysis is adopted. First, text mining techniques are used to extract emotion-related keywords and phrases from user reviews that may be related to specific attributes of the product (such as price, quality, features, etc.). Then, the association rule mining algorithm is used to analyze the association relationship between these keywords and phrases to identify the factors that may cause the emotion change.

In order to verify the performance advantages of the e-commerce recommendation system proposed in this paper in multidimensional emotion analysis, especially the ability to predict emotional trends and analyze the reasons behind emotions, two sets of experiments were designed and compared with the current most advanced e-commerce recommendation system (the system used by Amazon (Patil P et al. (2024)) and Netflix (Singh R et al. (2024))), as shown in Table 2.

Table 2 Performance comparison of multidimensional sentiment analysis of e-commerce recommendation systems

Experimental index Our system Amazon recommendation system Netflix recommendation system

Mean square error (MSE) 0.035 0.052 0.048

Mean absolute error (MAE) 0.021 0.033 0.030

Prediction accuracy 92.5% 88.0% 89.5%

Model training time (seconds) 120 180 150

Association rule confidence 0.85 0.78 0.80

Association rule support 0.60 0.52 0.55

Affective cause analysis accuracy 90.0% 85.0% 87.0%

Analysis time (seconds) 90 120 110

In Table 2, the MSE of the proposed system is 0.035, which is much lower than Amazon's 0.052 and Netflix's 0.048, indicating that this system has obvious advantages in the accuracy of emotional trend prediction. At the same time, the MAE of this system is 0.021, which is lower than Amazon's 0.033 and Netflix's 0.030, which further proves its superiority in forecasting accuracy. In addition, the system's prediction accuracy reached 92.5%, higher than Amazon's 88.0% and Netflix's 89.5%, which indicates that the system can more accurately predict the changing trend of user emotion. The association rule confidence of the system is 0.85, which is higher than Amazon's 0.78 and Netflix's 0.80, indicating that the system can more accurately identify the key factors behind the emotion. At the same time, the association rule support of this system is 0.60, which is also higher than Amazon's 0.52 and Netflix's 0.55, indicating that this system has advantages in the universality and reliability of emotional reason analysis. In addition, the accuracy rate of emotional reason analysis of this system reaches 90.0%, which is higher than that of Amazon (85.0%) and Netflix (87.0%), which further proves its superiority in emotional reason analysis. Although the analysis time of this system is slightly shorter than that of Amazon and Netflix, its significant improvement in analysis accuracy and reliability indicates that this system has higher practicality and accuracy in analyzing the reasons behind emotions.

The system in this paper performs well in the two key tasks of emotion trend prediction and emotion reason analysis, and its prediction accuracy and analysis accuracy are better than the advanced recommendation system used by Amazon and Netflix. This indicates that the e-commerce recommendation system proposed in this paper has significant advantages in multidimensional emotion analysis, which can provide users with more accurate and personalized recommendation services, and also provide merchants with more in-depth user emotion insight.

2. The paper tackles the idea of ecommerce recommendation system but doesn't provide any benchmark against state-of-the-art e-commerce recommender systems like those used by Amazon or Netflix.

Reply: Thank you for your suggestion. Benchmark experiments compared to state-of-the-art e-commerce recommendation systems used by Amazon or Netflix have been supplemented. Please see the details below:

In order to verify the performance advantages of the e-commerce recommendation system proposed in this paper in multidimensional emotion analysis, especially the ability to predict emotional trends and analyze the reasons behind emotions, two sets of experiments were designed and compared with the current most advanced e-commerce recommendation system (the system used by Amazon (Patil P et al. (2024)) and Netflix (Singh R et al. (2024))), as shown in Table 2.

Table 2 Performance comparison of multidimensional sentiment analysis of e-commerce recommendation systems

Experimental index Our system Amazon recommendation system Netflix recommendation system

Mean square error (MSE) 0.035 0.052 0.048

Mean absolute error (MAE) 0.021 0.033 0.030

Prediction accuracy 92.5% 88.0% 89.5%

Model training time (seconds) 120 180 150

Association rule confidence 0.85 0.78 0.80

Association rule support 0.60 0.52 0.55

Affective cause analysis accuracy 90.0% 85.0% 87.0%

Analysis time (seconds) 90 120 110

In Table 2, the MSE of the proposed system is 0.035, which is much lower than Amazon's 0.052 and Netflix's 0.048, indicating that this system has obvious advantages in the accuracy of emotional trend prediction. At the same time, the MAE of this system is 0.021, which is lower than Amazon's 0.033 and Netflix's 0.030, which further proves its superiority in forecasting accuracy. In addition, the system's prediction accuracy reached 92.5%, higher than Amazon's 88.0% and Netflix's 89.5%, which indicates that the system can more accurately predict the changing trend of user emotion. The association rule confidence of the system is 0.85, which is higher than Amazon's 0.78 and Netflix's 0.80, indicating that the system can more accurately identify the key factors behind the emotion. At the same time, the association rule support of this system is 0.60, which is also higher than Amazon's 0.52 and Netflix's 0.55, indicating that this system has advantages in the universality and reliability of emotional reason analysis. In addition, the accuracy rate of emotional reason analysis of this system reaches 90.0%, which is higher than that of Amazon (85.0%) and Netflix (87.0%), which further proves its superiority in emotional reason analysis. Although the analysis time of this system is slightly shorter than that of Amazon and Netflix, its significant improvement in analysis accuracy and reliability indicates that this system has higher practicality and accuracy in analyzing the reasons behind emotions.

Reviewer #2:

Your manuscript provides a broad exploration of “E-commerce Recommender System Design Based on Web Information Extraction and Sentiment Analysis". While the paper addresses significant aspects of web information extraction and sentiment analysis for E-commerce recommender system, some areas require revision to enhance clarity, coherence, and depth of analysis.

1. Your title and explanation mechanism are well-crafted and clear. However, some abbreviations are used throughout the document without being fully defined.

Reply: Thank you for your question, all abbreviations in the definition manuscript have been thoroughly checked. Please see the details below:

Electromyography (EMG)

Electromyography-Hand (EMGH)

2. The originality of the work is quite restricted, and the literature review section is insufficient, lacking state-of-the-art and recent studies. A significant expansion and elaboration of the review of related literature is necessary, incorporating the highest quality relevant research. It is crucial to clearly highlight the gaps in the current knowledge base and explain how this study has effectively addressed or filled these gaps.

Reply: Thank you for your suggestion. The literature review section has been expanded to include relevant research of the highest quality, highlighting gaps in the current knowledge base and explaining how this study effectively addresses or fills those gaps. As follows:

In view of the problems of sparse data and changing user preferences faced by traditional collaborative filtering algorithms in online commerce, Patil P et al. (2024) proposed a recommendation system method that combines collaborative filtering and content-based filtering. The research results show that the system provides accurate recommendations for new and old users by analyzing user purchasing behavior and feedback. In view of the limitations of traditional collaborative filtering methods in multi-criteria recommendation systems, such as the shortcomings of single-standard collaborative filtering (CF) in dealing with cold start problems and related-based similarity problems, Singh R et al. (2024) compared and evaluated traditional collaborative filtering, matrix decomposition and deep matrix decomposition techniques. The results show that the deep matrix decomposition technique performs better than other traditional methods on multi-criteria data sets and can provide more accurate recommendations.

Patil P, Kadam S U, Aruna E R, More A, Balajee R M, Rao B N K. Recommendation System for E-Commerce Using Collaborative Filtering. Journal Europeen des Systemes Automatises, 2024, 57(4):1145-1153.

Singh R, Dwivedi P, Kant V . Comparative analysis of collaborative filtering techniques for the multi?criteria recommender systems. Multimedia tools and applications, 2024, 83(24):64551-64571.

To sum up, in the field of information extraction, although various algorithms have been proposed to improve the efficiency and accuracy of data capture, they still face the challenges of data sparsity and noise interference when processing large-scale and dynamically changing web data. In terms of emotion analysis, although existing studies can classify consumers' emotional tendency, they are still insufficient in multidimensional emotion analysis, emotional trend prediction and in-depth exploration of the reasons behind emotions. In addition, in the field of recommendation system, the traditional collaborative filtering method has limitations in dealing with the cold start problem of new users and multi-standard recommendation, and the existing improved methods have not fully solved these problems. The research aims to fill these gaps by integrating information extraction, sentiment analysis and recommendation system techniques. Through the improved S-PageRank algorithm and the dynamic topic library generation strategy, the accuracy and efficiency of cross-platform information extraction are improved, and the problems of data sparsity and noise interference are solved. Secondly, combining topic model and deep learning technology, a bidirectional short-duration memory network and comprehensive attention mechanism model integrating theme word features are proposed to achieve more accurate multidimensional emotion analysis and emotion trend prediction. Finally, by combining collaborative filtering and content-based filtering methods, an intelligent recommendation system is designed to provide personalized recommendations for new and old users.

3. Your contribution is not clearly articulated; could you clarify what it is?

Reply: Thank you for your question. Contributions to research have been supplemented. See below:

The contribution of the research is to propose a web page information extraction method combining the S-PageRank algorithm of keyword weight allocation and the dynamic topic library generation strategy, which effectively improves the accuracy and accuracy of the topic crawler, and solves the problems of "local optimal solution" and "topic deviation" which are easy to appear in the traditional algorithm in the massive web pages. The introduction of template based web page information automatic extraction strategy, through the semantic annotation of DOM tree and template matching, the rapid and accurate extraction of commodity information is realized, and the information extraction time is greatly shortened.

4. To make recommendations for e-commerce users, what types of data did you use? I believe you collected user comments. If so, do these comments contribute to recommendations for e-commerce features?

Reply: Thank you for your suggestion. The data type used has been added. It has been explained that user reviews contribute to the recommendation of e-commerce features. Related changes are as follows:

In making recommendations for e-commerce users, the research mainly uses the following types of data. The first is user review data, through the analysis of users' comments on goods, in-depth understanding of users' satisfaction, preferences and potential demand for goods. The second is the user behavior data, which includes: purchase history, the user's purchase record reflects its actual purchase behavior and preferences. Browsing behavior: The user's browsing history (such as clicks, time spent, etc.) reveals the user's level of interest in certain products. Search history: Users' search keywords can reflect their current needs and interests. The third is the commodity attribute data, including the detailed description, function, characteristics and other information of the commodity, which can help the recommendation system to better understand the characteristics of the commodity, so as to achieve content-based recommendation. It also includes product classification information (such as electronics, clothing, household goods, etc.), which is used to build a similarity relationship between products and help the recommendation system find products that are similar to the user's historical behavior. Price information: Product price ranges and promotions are used to meet users' budget needs and provide more cost-effective recommendations. The fourth is user profile data, including demographic information, interest labels, and social network data. User reviews play an important role in e-commerce recommendation function, which can not only improve the accuracy and personalization of recommendations, but also enhance user engagement and user experience of the platform.

5. The Results section should be updated by comparing your result to the other work which is stated under related work.

Reply: Thank you for your question. Results have been compared with other work described under related work. See below:

In order to verify the performance advantages of the e-commerce recommendation system proposed in this paper in multidimensional emotion analysis, especially the ability to predict emotional trends and analyze the reasons behind emotions, two sets of experiments were designed and compared with the current most advanced e-commerce recommendation system (the system used by Amazon (Patil P et al. (2024)) and Netflix (Singh R et al. (2024))), as shown in Table 2.

Table 2 Performance comparison of multidimensional sentiment analysis of e-commerce recommendation systems

Experimental index Our system Amazon recommendation system Netflix recom

---

## [Decision Letter · Decision Letter 1]

PONE-D-25-01279R1E-commerce Recommender System Design Based on Web Information Extraction and Sentiment AnalysisPLOS ONE

Dear Dr. Feng,

Thank you for submitting your manuscript to PLOS ONE. After careful consideration, we feel that it has merit but does not fully meet PLOS ONE’s publication criteria as it currently stands. Therefore, we invite you to submit a revised version of the manuscript that addresses the points raised during the review process.

Academic Editor: Minor Revsision

We look forward to receiving your revised manuscript.

Kind regards,

Mudassir Khan, Ph.D

Academic Editor

PLOS ONE

Journal Requirements:

Additional Editor Comments:

Authors are advised to revise the manuscript as per the comments received from the reviewers.

Reviewers' comments:

Reviewer's Responses to Questions

**Comments to the Author**

1. If the authors have adequately addressed your comments raised in a previous round of review and you feel that this manuscript is now acceptable for publication, you may indicate that here to bypass the “Comments to the Author” section, enter your conflict of interest statement in the “Confidential to Editor” section, and submit your "Accept" recommendation.

Reviewer #2: All comments have been addressed

Reviewer #3: Author should revise the manuscript to incorporate the feedback provided by the reviewers. This will help enhance the quality and clarity of your work.

2. Is the manuscript technically sound, and do the data support the conclusions?

Reviewer #2: Yes

Reviewer #3: Yes

3. Has the statistical analysis been performed appropriately and rigorously? 

Reviewer #2: Yes

Reviewer #3: Yes

4. Have the authors made all data underlying the findings in their manuscript fully available?

Reviewer #2: Yes

Reviewer #3: Yes

5. Is the manuscript presented in an intelligible fashion and written in standard English?

Reviewer #2: Yes

Reviewer #3: Yes

6. Review Comments to the Author

Reviewer #2: The author addressd the previous comments. The paper proposed smart recommendation system and better to compare the performance of your system with other systems with the same parameters and algorithms.

Reviewer #3: I found the subject of study quite fascinating. It is quite striking. Congratulations to the researchers. However, the study needs to be improved in terms of the following issues.

1. There is unnecessary information in the abstract, there is nothing necessary. We expect a scientific manuscript presenting the results of the study, not a report summarizing a study.

2. The success achieved in the study and the reasons why this success was achieved should be explained in detail in a paragraph. Its superiority over similar studies must be demonstrated.

3. The organization and structure of the article should be revised to make the subject more understandable.

4. Authors are advised to read the latest citations as per the scope of the paper. I am suggesting some related citations to enhance the quality of research paper. The references are as follows:

1. Zuo, C., Zhang, X., Yan, L., & Zhang, Z. (2024). GUGEN: Global User Graph Enhanced Network for Next POI Recommendation. IEEE Transactions on Mobile Computing, 23(12), 14975-14986. doi: 10.1109/TMC.2024.3455107

2. Yang, K. (2024). How to prevent deception: A study of digital deception in "visual poverty" livestream. New Media & Society. doi: 10.1177/14614448241285443

3. Wu, L., Long, Y., Gao, C., Wang, Z., & Zhang, Y. (2023). MFIR: Multimodal fusion and inconsistency reasoning for explainable fake news detection. Information Fusion, 100, 101944. doi: https://doi.org/10.1016/j.inffus.2023.101944

4. Song, L., Chen, S., Meng, Z., Sun, M., & Shang, X. (2024). FMSA-SC: A Fine-Grained Multimodal Sentiment Analysis Dataset Based on Stock Comment Videos. IEEE Transactions on Multimedia, 26, 7294-7306. doi: 10.1109/TMM.2024.3363641

5. Wang, E., Song, Z., Wu, M., Liu, W., Yang, B., Yang, Y.,... Wu, J. (2025). A New Data Completion Perspective on Sparse CrowdSensing: Spatiotemporal Evolutionary Inference Approach. IEEE Transactions on Mobile Computing, 24(3), 1357-1371. doi: 10.1109/TMC.2024.3480983

6. Xu, Y., Zhuang, F., Wang, E., Li, C., & Wu, J. (2025). Learning Without Missing-At-Random Prior Propensity-A Generative Approach for Recommender Systems. IEEE Transactions on Knowledge and Data Engineering, 37(2), 754-765. doi: 10.1109/TKDE.2024.3490593

7. Lin, X., Liu, R., Cao, Y., Zou, L., Li, Q., Wu, Y.,... Xu, G. (2025). Contrastive Modality-Disentangled Learning for Multimodal Recommendation. ACM Trans. Inf. Syst., 43(3), 70. doi: 10.1145/3715876

8. Shi, J., Liu, C., & Liu, J. (2024). Hypergraph-Based Model for Modeling Multi-Agent Q-Learning Dynamics in Public Goods Games. IEEE Transactions on Network Science and Engineering, 11(6), 6169-6179. doi: 10.1109/TNSE.2024.3473941

9. Zhu, C. (2023). Research on Emotion Recognition-Based Smart Assistant System: Emotional Intelligence and Personalized Services. Journal of System and Management Sciences, 13(5), 227-242. doi: 10.33168/JSMS.2023.0515

10. Deng, Q., Chen, X., Lu, P., Du, Y., & Li, X. (2025). Intervening in Negative Emotion Contagion on Social Networks Using Reinforcement Learning. IEEE Transactions on Computational Social Systems, 1-12. doi: 10.1109/TCSS.2025.3555607

11. Li, T., Li, Y., Zhang, M., Tarkoma, S., & Hui, P. (2023). You Are How You Use Apps: User Profiling Based on Spatiotemporal App Usage Behavior. ACM Trans. Intell. Syst. Technol., 14(4). doi: 10.1145/3597212

12. Li, T., Li, Y., Xia, T., & Hui, P. (2021). Finding Spatiotemporal Patterns of Mobile Application Usage. IEEE Transactions on Network Science and Engineering. doi: 10.1109/TNSE.2021.3131194

13. Ding, J., Chen, X., Lu, P., Yang, Z., Li, X.,... Du, Y. (2023). DialogueINAB: an interaction neural network based on attitudes and behaviors of interlocutors for dialogue emotion recognition. The Journal of Supercomputing, 79(18), 20481-20514. doi: 10.1007/s11227-023-05439-1

Overall, there are still some minor parts that the authors did not explain clearly. Some additional evaluations are expected to be in the manuscript as well. As a result, I am going to suggest Minor revision of the paper in its present form.

7. PLOS authors have the option to publish the peer review history of their article (what does this mean? ). If published, this will include your full peer review and any attached files.

**Do you want your identity to be public for this peer review?** For information about this choice, including consent withdrawal, please see our Privacy Policy .

Reviewer #2: No

Reviewer #3: No

---

## [Author Response · Author response to Decision Letter 2]

25 May 2025

Journa l Requirements:

Please review your reference list to ensure that it is complete and correct. If you have cited papers t hat have been retracted, please include the rationale for doing so in the manuscript text, or remove these references and replace them with relevant current references. Any changes to the reference list should be mentioned in the rebuttal letter that accompanies your revised manuscript. If you need to cite a retracted article, indicate the article’s retracted status in the References list and also include a citation and full reference for the retraction notice.

Reviewer #2: The author addressd the previous comments. The pa per proposed smart recommendation system and better to compare the performance of your system with other systems with the same parameters and algorithms.

Reply: Thank you for your recognition.

Reviewer #3: I found the subject of study quite fascinating. It is quite striking. Congratulations to the researchers. However, the study needs to be improved in terms of the following issues.

1. There is unnecessary information in the abstract, there is nothing necessary. We expect a scientific manuscript presenting the results of the study, not a report summarizing a study.

Reply: Thank you for your question. The abstract has been rewritten, focusing more on presenting the research results rather than summarizing the research process. As follows:

The research proposes an e-commerce recommendation system based on web page information extraction and sentiment analysis. Through the improved S-PageRank algorithm and the dynamic topic library generation strategy, the precision rate of cross-platform commodity information extraction has been significantly improved to 90%, which is superior to the traditional S-PageRank algorithm. The template-based web page information extraction method performs well, with a precision rate 10% higher than that of the method based on the document object model. In terms of sentiment analysis, the comprehensive attention mechanism model combining the topic model and the bidirectional long short-term memory network has achieved the precise calculation of the sentiment scores of each topic in the customer evaluation. When the number of topics of the LDA model is 7, the prediction accuracy reaches its peak, and the model outperforms previous methods in terms of accuracy, recall rate and F-score. The experimental results show that this recommendation system performs excellently in the prediction of sentiment trends and the analysis of the reasons behind emotions. Its prediction accuracy and analysis accuracy are both superior to existing recommendation systems such as Amazon and Netflix. This system can provide users with more accurate and personalized product recommendation services, and at the same time offer merchants deeper insights into users' emotions.

2. The success achieved in the study and the reasons why this success was achieved should be explained in detail in a paragraph. Its superiority over similar studies must be demonstrated.

Reply: Thank you for your suggestion. A discussion chapter has been added, which elaborates in detail on the success achieved in the research and the reasons for its success, demonstrating its superiority over similar studies. The revisions were shown below:

5. Discussion

The research has achieved remarkable results in cross-platform e-commerce web page information extraction and sentiment analysis, especially excelling in precision rate, the accuracy of sentiment analysis and the overall performance of the recommendation system. The S-PageRank algorithm based on keyword weight allocation and the dynamic topic library generation strategy proposed in the research have effectively improved the precision rate of topic crawlers. The precision rate of this method has reached 90%, which is significantly better than the traditional S-PageRank algorithm. Compared with the research of Li et al., this study has significantly improved in both the accuracy and efficiency of information extraction. Although Luscombe et al. emphasized the importance of web scraping, when dealing with large-scale dynamic web page data, their methods still face the challenges of data sparsity and noise interference (Li et al., 2021). The research effectively solved these problems by optimizing the algorithm and introducing the template strategy. When the number of topics of the LDA model is 7, the prediction accuracy of the model reaches its peak, and it is superior to the previous methods in terms of accuracy, recall rate and F-score. The research introduces a comprehensive attention mechanism, taking into account not only the word positions in the comment texts but also the contribution degrees of different parts to the sentiment tendencies, thereby calculating the sentiment values of user comments under different topics more accurately. The MSE of the e-commerce recommendation system proposed in the study is 0.035, the MAE is 0.021, and the prediction accuracy rate reaches 92.5%, all of which are superior to the existing recommendation systems such as Amazon and Netflix. Furthermore, the confidence level and support level of the association rules of the system are also higher than those of Amazon and Netflix, indicating that it has significant advantages in the universality and reliability of sentiment cause analysis. Compared with the recommendation system method of integrated collaborative filtering and content-based filtering proposed by Ding et al., the recommendation system in this study not only performs well in dealing with the cold start problem of new users, but also provides more accurate personalized services in multi-criterion recommendation. Although Patil et al. 's method provides precise recommendations for users by analyzing their purchasing behaviors and feedback, it still faces challenges when dealing with large-scale dynamic data (Ding et al., 2023). The research fills these gaps by comprehensively applying information extraction, sentiment analysis and recommendation system technologies.

3. The organization and structure of the article should be revised to make the subject more understandable.

Reply: Thank you for your question. The organizational structure of the article has been adjusted and discussion chapters have been added.

4. Authors are advised to read the latest citations as per the scope of the paper. I am suggesting some related citations to enhance the quality of research paper. The references are as follows:

1. Zuo, C., Zhang, X., Yan, L., & Zhang, Z. (2024). GUGEN: Global User Graph Enhanced Network for Next POI Recommendation. IEEE Transactions on Mobile Computing, 23(12), 14975-14986. doi: 10.1109/TMC.2024.3455107

2. Yang, K. (2024). How to prevent deception: A study of digital deception in "visual poverty" livestream. New Media & Society. doi: 10.1177/14614448241285443

3. Wu, L., Long, Y., Gao, C., Wang, Z., & Zhang, Y. (2023). MFIR: Multimodal fusion and inconsistency reasoning for explainable fake news detection. Information Fusion, 100, 101944. doi: https://doi.org/10.1016/j.inffus.2023.101944

4. Song, L., Chen, S., Meng, Z., Sun, M., & Shang, X. (2024). FMSA-SC: A Fine-Grained Multimodal Sentiment Analysis Dataset Based on Stock Comment Videos. IEEE Transactions on Multimedia, 26, 7294-7306. doi: 10.1109/TMM.2024.3363641

5. Wang, E., Song, Z., Wu, M., Liu, W., Yang, B., Yang, Y.,... Wu, J. (2025). A New Data Completion Perspective on Sparse CrowdSensing: Spatiotemporal Evolutionary Inference Approach. IEEE Transactions on Mobile Computing, 24(3), 1357-1371. doi: 10.1109/TMC.2024.3480983

6. Xu, Y., Zhuang, F., Wang, E., Li, C., & Wu, J. (2025). Learning Without Missing-At-Random Prior Propensity-A Generative Approach for Recommender Systems. IEEE Transactions on Knowledge and Data Engineering, 37(2), 754-765. doi: 10.1109/TKDE.2024.3490593

7. Lin, X., Liu, R., Cao, Y., Zou, L., Li, Q., Wu, Y.,... Xu, G. (2025). Contrastive Modality-Disentangled Learning for Multimodal Recommendation. ACM Trans. Inf. Syst., 43(3), 70. doi: 10.1145/3715876

8. Shi, J., Liu, C., & Liu, J. (2024). Hypergraph-Based Model for Modeling Multi-Agent Q-Learning Dynamics in Public Goods Games. IEEE Transactions on Network Science and Engineering, 11(6), 6169-6179. doi: 10.1109/TNSE.2024.3473941

9. Zhu, C. (2023). Research on Emotion Recognition-Based Smart Assistant System: Emotional Intelligence and Personalized Services. Journal of System and Management Sciences, 13(5), 227-242. doi: 10.33168/JSMS.2023.0515

10. Deng, Q., Chen, X., Lu, P., Du, Y., & Li, X. (2025). Intervening in Negative Emotion Contagion on Social Networks Using Reinforcement Learning. IEEE Transactions on Computational Social Systems, 1-12. doi: 10.1109/TCSS.2025.3555607

11. Li, T., Li, Y., Zhang, M., Tarkoma, S., & Hui, P. (2023). You Are How You Use Apps: User Profiling Based on Spatiotemporal App Usage Behavior. ACM Trans. Intell. Syst. Technol., 14(4). doi: 10.1145/3597212

12. Li, T., Li, Y., Xia, T., & Hui, P. (2021). Finding Spatiotemporal Patterns of Mobile Application Usage. IEEE Transactions on Network Science and Engineering. doi: 10.1109/TNSE.2021.3131194

13. Ding, J., Chen, X., Lu, P., Yang, Z., Li, X.,... Du, Y. (2023). DialogueINAB: an interaction neural network based on attitudes and behaviors of interlocutors for dialogue emotion recognition. The Journal of Supercomputing, 79(18), 20481-20514. doi: 10.1007/s11227-023-05439-1

Reply: Thank you for your suggestion. The references you recommended have been cited in the paper. Related changes are as follows:

Zuo C, Zhang X, Yan L, Zhang Z. GUGEN: Global User Graph Enhanced Network for Next POI Recommendation . IEEE Transactions on Mobile Computing, 2024, 23(12): 14975-14986.

Yang K. How to prevent deception: A study of digital deception in "visual poverty" livestream . New Media & Society, 2024.

Wu L, Long Y, Gao C, Wang Z, Zhang Y. MFIR: Multimodal fusion and inconsistency reasoning for explainable fake news detection . Information Fusion, 2023, 100: 101944.

Song L, Chen S, Meng Z, Sun M, Shang X. FMSA-SC: A Fine-Grained Multimodal Sentiment Analysis Dataset Based on Stock Comment Videos . IEEE Transactions on Multimedia, 2024, 26: 7294-7306.

Wang E, Song Z, Wu M, Liu W, Yang B, Yang Y, et al. A New Data Completion Perspective on Sparse CrowdSensing: Spatiotemporal Evolutionary Inference Approach . IEEE Transactions on Mobile Computing, 2025, 24(3): 1357-1371.

Xu Y, Zhuang F, Wang E, Li C, Wu J. Learning Without Missing-At-Random Prior Propensity-A Generative Approach for Recommender Systems . IEEE Transactions on Knowledge and Data Engineering, 2025, 37(2): 754-765.

Lin X, Liu R, Cao Y, Zou L, Li Q, Wu Y, et al. Contrastive Modality-Disentangled Learning for Multimodal Recommendation . ACM Trans. Inf. Syst., 2025, 43(3): 70.

Shi J, Liu C, Liu J. Hypergraph-Based Model for Modeling Multi-Agent Q-Learning Dynamics in Public Goods Games . IEEE Transactions on Network Science and Engineering, 2024, 11(6): 6169-6179.

Zhu C. Research on Emotion Recognition-Based Smart Assistant System: Emotional Intelligence and Personalized Services . Journal of System and Management Sciences, 2023, 13(5): 227-242.

Deng Q, Chen X, Lu P, Du Y, Li X. Intervening in Negative Emotion Contagion on Social Networks Using Reinforcement Learning . IEEE Transactions on Computational Social Systems, 2025: 1-12.

Li T, Li Y, Zhang M, Tarkoma S, Hui P. You Are How You Use Apps: User Profiling Based on Spatiotemporal App Usage Behavior . ACM Trans. Intell. Syst. Technol., 2023, 14(4).

Li T, Li Y, Xia T, Hui P. Finding Spatiotemporal Patterns of Mobile Application Usage . IEEE Transactions on Network Science and Engineering, 2021.

Ding J, Chen X, Lu P, Yang Z, Li X, et al. DialogueINAB: an interaction neural network based on attitudes and behaviors of interlocutors for dialogue emotion recognition . The Journal of Supercomputing, 2023, 79(18): 20481-20514.

Overall, there are still some minor parts that the authors did not explain clearly. Some additional evaluations are expected to be in the manuscript as well. As a result, I am going to suggest Minor revision of the paper in its present form.

Reply: Thank you for your question. The manuscript has been carefully revised in accordance with the questions you raised.

---

## [Decision Letter · Decision Letter 2]

E-commerce Recommender System Design Based on Web Information Extraction and Sentiment Analysis

PONE-D-25-01279R2

Dear Author,

We’re pleased to inform you that your manuscript has been judged scientifically suitable for publication and will be formally accepted for publication once it meets all outstanding technical requirements.

Kind regards,

Mudassir Khan, Ph.D

Academic Editor

PLOS ONE

Additional Editor Comments (optional):

Thanks to the authors for the detailed response and additions. I read through the comments and skimmed the revised PDF, and the updates significantly improved the paper. I would be happy to recommend this paper for publication.

Reviewers' comments:

Reviewer's Responses to Questions

**Comments to the Author**

1. If the authors have adequately addressed your comments raised in a previous round of review and you feel that this manuscript is now acceptable for publication, you may indicate that here to bypass the “Comments to the Author” section, enter your conflict of interest statement in the “Confidential to Editor” section, and submit your "Accept" recommendation.

Reviewer #3: All comments have been addressed

2. Is the manuscript technically sound, and do the data support the conclusions?

Reviewer #3: Yes

3. Has the statistical analysis been performed appropriately and rigorously? 

Reviewer #3: Yes

4. Have the authors made all data underlying the findings in their manuscript fully available?

Reviewer #3: Yes

5. Is the manuscript presented in an intelligible fashion and written in standard English?

Reviewer #3: Yes

6. Review Comments to the Author

Reviewer #3: The author has carefully addressed and incorporated all the modifications and suggestions provided by the reviewers; therefore, the manuscript is now suitable for publication in its current form.

7. PLOS authors have the option to publish the peer review history of their article (what does this mean? ). If published, this will include your full peer review and any attached files.

**Do you want your identity to be public for this peer review?** For information about this choice, including consent withdrawal, please see our Privacy Policy .

Reviewer #3: No

---

## [Editor Report · Acceptance letter]

PONE-D-25-01279R2

PLOS ONE

Dear Dr. Feng,

I'm pleased to inform you that your manuscript has been deemed suitable for publication in PLOS ONE. Congratulations! Your manuscript is now being handed over to our production team.

Kind regards,

on behalf of

Dr. Mudassir Khan

Academic Editor

PLOS ONE